# Kynurenic acid, a key L-tryptophan-derived metabolite, protects the heart from an ischemic damage

Einat Bigelman[1,2], Metsada Pasmanik-Chor[3], Bareket Dassa[4], Maxim Itkin[5], Sergey Malitsky[5], Orly Dorot[6], Edward Pichinuk[6], Yuval Kleinberg[1,6], Gad Keren[1,2], Michal Entin-Meer[1,2]*

1 Laboratory of Cardiovascular Research, Tel Aviv Sourasky Medical Center, Affiliated with the Sackler Faculty of Medicine, Tel-Aviv University, Tel-Aviv, Israel, 2 Department of Cardiology, Sackler School of Medicine, Tel-Aviv University, Tel-Aviv, Israel, 3 Bioinformatics Unit, Faculty of Life Sciences, Tel-Aviv University, Tel-Aviv, Israel, 4 Bioinformatics Unit, Department of Life Sciences Core Facilities, Weizmann Institute of Science, Rehovot, Israel, 5 Metabolic Profiling Unit, Life Sciences Core Facilities, Weizmann Institute of Science, Rehovot, Israel, 6 Bio-Imaging Core, Blavatnik Center for Drug Discovery, Tel-Aviv University, Tel-Aviv, Israel

* michale@tlvmc.gov.il

**Data Availability Statement:** All relevant data are within the paper and its Supporting Information files.

## Abstract

### Background

Renal injury induces major changes in plasma and cardiac metabolites. Using a small- animal *in vivo* model, we sought to identify a key metabolite whose levels are significantly modified following an acute kidney injury (AKI) and to analyze whether this agent could offer cardiac protection once an ischemic event has occurred.

### Methods and results

Metabolomics profiling of cardiac lysates and plasma samples derived from rats that underwent AKI 1 or 7 days earlier by 5/6 nephrectomy versus sham-operated controls was performed. We detected 26 differential metabolites in both heart and plasma samples at the two selected time points, relative to sham. Out of which, kynurenic acid (kynurenate, KYNA) seemed most relevant. Interestingly, KYNA given at 10 mM concentration significantly rescued the viability of H9C2 cardiac myoblast cells grown under anoxic conditions and largely increased their mitochondrial content and activity as determined by flow cytometry and cell staining with MitoTracker dyes. Moreover, KYNA diluted in the drinking water of animals induced with an acute myocardial infarction, highly enhanced their cardiac recovery according to echocardiography and histopathology.

### Conclusion

KYNA may represent a key metabolite absorbed by the heart following AKI as part of a compensatory mechanism aiming at preserving the cardiac function. KYNA preserves the *in vitro* myocyte viability following exposure to anoxia in a mechanism that is mediated, at least in part, by protection of the cardiac mitochondria. A short-term administration of KYNA may

**Funding:** GK, Azrieli Research Fund GK, 212843 Ichilov-Weizmann Research Fund MEM and GK, Ziternik Fund, Tel-Aviv University The funders had no role in the study design, data collection and analysis, decision to publish, or preparation of the manuscript.

**Competing interests:** The authors have declared that no competing interests exist.

be highly beneficial in the treatment of the acute phase of kidney disease in order to attenuate progression to reno-cardiac syndrom and to reduce the ischemic myocardial damage following an ischemic event.

## Introduction

Concomitant cardiac and renal dysfunction, termed cardiorenal syndrome (CRS), has gained significant attention in the recent decade and is characterized into 5 types [1]. Type 3 and type 4 CRS, also known as renocardiac syndrome, are defined by acute or chronic kidney disease (AKI and CKD, respectively) and are associated with the development or progression of cardiac disease [2]. Kidney disease may directly or indirectly produce an acute cardiac event triggered by the inflammatory surge, oxidative stress and secretion of neurohormons [3–6]. Other known triggers for mainly AKI- mediated cardiac injuries and dysfunctions include AKI-related volume overload, metabolic acidosis and electrolyte disorders such as hyperkalemia and hypocalcemia [7]. Consequently, renal disease is considered a major risk for cardiovascular complications including acute myocardial infarction (AMI) and congestive heart failure. Recently, changes in cardiac mitochondrial dynamics and subsequent cardiac apoptosis have been suggested in a mouse renal ischemia-reperfusion (I/R) model. In this study, fragmented mitochondria in cardiomyocytes were observed after 30 min of bilateral renal I/R and reduction of fractional shortening was observed 72 h later [8]. Likewise, in our recent work, we have shown that sustained CKD induced in a rat model by 5/6 nephrectomy results in major cardiac pathology, including increased interstitial fibrosis, cardiomyocyte hypertrophy, induced expression of pro-apoptotic markers and massive spatial disarrangement of the cardiac muscle fibers. These changes were combined with significant mitochondrial damage as reflected by the swollen mitochondria in which the cristae density was reduced [9]. The processes were associated with an induced expression of the fission-related protein DRP1, suggesting that cardiac mitochondrial fragmentation takes place upon kidney injury [8, 9]. Interestingly, a recent study has demonstrated that AKI induces changes in 40% of the cardiac metabolites [10]. In this study, the authors documented that the post-AKI cardiac metabolome was characterized by amino acid depletion, increased oxidative stress, and evidence of alternative energy production. In view of these data, in the current work, we wished to identify a key metabolite pathway whose expression is highly modified in both blood and cardiac lysates of Lewis rats induced with AKI in order to characterize a potential novel interventional tool for attenuating disease progression to CRS. Thorough metabolomics analysis revealed major changes in the L-tryptophan (TRP) metabolism pathway in general and a significant induction in kynurenic acid (kynurenate, KYNA), one of the major products of the TRP metabolism both in the blood and hearts of the experimental rats. Therefore we chose to focus in the current work on KYNA's effects on cardiac structure and function. Several lines of evidence suggest that G protein-coupled receptor 35 (GPR35) may serve as KYNA's receptor and that GPR35 can be activated and internalized into the cell by KYNA, which is found in the plasma in nano to micromolar concentrations [11]. Knock-out mice lacking GPR35 expression have been reported to have increased systemic blood pressure [12], but the mechanism is not known. In addition, high amounts of GPR35 were shown to be expressed in the wild-type mice hearts, and myocardial GPR35 gene expression was shown to be associated with human heart failure [13, 14].

In the current study, we analyzed the potential effects of KYNA on cardiac cell viability following exposure to ischemia using *in vitro* and *in vivo* models and assessed the possible

involvement of the cardiac mitochondria in mediating the metabilte's effects. The data presented herein highlight the potential crucial importance of KYNA as a key metabolite that may offer cardiac protection under an ischemic event prevalent in patients suffering from renal dysfunction.

## Materials & methods

### Ethics statement

Animal studies were approved by The Animal Care and Use Committee of Tel Aviv Sourasky Medical-Center (6-1-18), which conforms to the policies of the American Heart Association and the Guide for the Care and Use of Laboratory Animals.

### Animals

Male Lewis rats (250–350 gr)/ 12-week old female BALB/C mice were purchased from Envigo, Israel. Male Lewis rats were used for AKI studies, as previously established in our laboratory for 5/6 nephrectomy models [9]. Female mice were utilized in the AMI studies, in accordance with the extensive experience gathered in our laboratory [15, 16] and the lower cardiac rupture rate of female compared to male mice following AMI induction [17]. All animals were kept under optimal conditions (food and water provided *ad libitum*) at room temperature in a temperature-controlled facility with 12 hrs. light/dark cycle. Prior to all surgical procedures, animals were anesthetized with a mixture of ketamine/xylazine (100/10 mg/kg, respectively) and anesthesia was confirmed by loss of pedal reflex (toe pinch). During experimental periods, animals were monitored 2–3 times per week for potential signs of suffering, mainly weight loss of more than 15% and significant changes in animals' behavior, mobility, or body posture. Should animals have met one of these criteria, euthanasia would have been warranted on the same day to prevent further suffering.

### Rat model for AKI

Rats (4-5/group) underwent 5/6 nephrectomy for induction of AKI as previously established in our laboratory [9]. Briefly, 5/6 nephrectomy was performed in two consecutive surgeries: two-thirds of the left kidney was removed followed by removal of the right kidney a week later. Control animals underwent abdominal opening only (sham). The experiment was terminated one day or seven days following the second surgery. AKI-1d represents a potentially-transient disease as opposed to the 1w AKI that may represent a potentially- persistent acute renal disease.

### Mice model for AMI

Twelve week-old BALB/C female mice were randomized into two groups receiving tap water (control) or KYNA (250 mg/L) diluted in tap water (15 mice/arm) in accordance with previous reports [18]. Three days later all animals were induced with AMI by permanent ligation of the left anterior descending artery (LAD) according to our established protocol [16]. To alleviate pain, subcutaneous injection of carprofen (5 mg/kg) was given during the surgery as well as once a day during the next three subsequent days. Five animals from the control group and six animals from the experimental group were excluded from the analysis due to one of the following reasons: normal cardiac function according to echocardiography applied on day 1, suggesting that LAD ligation was unsuccessful (6 animals), animal's death prior to endpoint (4 animals) or weight loss of > 15% (1 animal). Consequently, the control and the experimental arms included 10 and 9 animals, respectively. The drinking water with or without KYNA was

replaced twice a week until the end of the experiment on day 30 post surgery. Cardiac parameters on day one and day 30 post-infarction were obtained by echocardiography (Vevo 2100, VisualSonics, Toronto, Canada), while heart rate was kept between 400–500 bpm in accordance with the guidelines for echocardiographic measurements in the murine heart [19]. Stroke volume (SV) was calculated as the difference the LV end-diastolic and end-systolic volumes (LVEDV and LVESV) which were automatically calculated by the Vevo cardiac software. The cardiac output (CO) was derived from the SV and heart rate. Following the second echocardiography scan, mice were euthanized by $CO_2$ inhalation, and hearts were harvested and processed for histology.

## Histological staining for collagen content

After embedding in paraffin, the blocks were sectioned into 5 μm slices and stained with Picro Sirius Red (PR, Direct Red 80, Sigma-Aldrich, St Louis, MO, USA) and hematoxylin to determine hypertrophy and the extent of left ventricular (LV) collagen scar.

## Metabolites extraction and profiling

The heart (LV) tissue was frozen in liquid nitrogen and then ground to powder using a mortar and pestle. The blood was withdrawn from the vena cava into blood collection tubes (BD Vacutainer). Prior to termination, urine samples were collected. The tubes were centrifuged for two minutes at 4000 rpm, and the plasma samples were collected. Extraction and analysis of lipids and polar/semi-polar metabolites was performed as previously described [20–22] with some modifications. Plasma (100 μL) or ground lyophilized LV samples (20 mg) were extracted with 1 ml of a pre-cooled (−20˚C) homogenous methanol:methyl-tert-butyl-ether (MTBE) 1:3 (v/v) mixture, containing following internal standards. The tubes were vortexed and then sonicated for 30 min in an ice-cold sonication bath (taken for a brief vortex every 10 min). Then, UPLC-grade water- methanol (3:1, v/v) solution (0.5 mL) was added to the tubes, followed by centrifugation. The upper organic phase was transferred into 2 mL Eppendorf tube. The polar phase was re-extracted as described above with 0.5 mL of MTBE. Both organic phases were combined and dried in speedvac and then stored at −80˚C until analysis.

## H9C2 cell viability assay

H9C2 myoblast cell line from rat myocardium (ATCC clone CRL-1446) was obtained as a generous gift from Dr. Gania Kessler-Icekson, Felsenstein Medical Research Center, Petach Tikva, Israel. The cells were plated in 6 well plates (2.5 $*10^5$ cells/ well) in complete DMEM. A day later, the medium was replaced with serum-free DMEM with or without KYNA. Twenty-four hours later, the cells were exposed to normoxic or anoxic (0% oxygen) conditions for additional 48 hours by applying a mixture of 95% $N_2$ + 5% $CO_2$ into a modular incubator chamber (Billups Rothenber), as previously reported [23]. The cells were then collected, and their viability was tested using the annexin-PI apoptosis detection kit (MBL, USA) followed by flow cytometry analysis (BD Biosciences FACS Canto II).

## Detection of cell surface GPR35

H9C2 were plated in 6 well plates (2.5 $*10^5$ cells/ well) in complete DMEM. A day later the medium was replaced with serum-free DMEM with or without KYNA followed by exposure to normoxia/anoxia for another 48 hours. The cells were then trypsinized and stained with GPR35-FITC antibody, which recognizes the native form of the extracellular portion of the receptor (Biorbyt, England, clone orb399781) followed by flow cytometry analysis.

## Cell staining and data analysis of mitochondrial morphology

H9C2 cells were plated in 96-well plates (10,000 cells/well). On the following day, the growth medium was replaced with serum-free DMEM, with or without KYNA. Following 24 hours, the plates were exposed to normoxia/anoxia for 48 hours. The cells were stained using an automated High Content Imaging pipline (Freedom EVO 200 robot, Tecan Group Ltd., Männedorf, Switzerland) and imaged using the INCell 2200 automated microscope (GE Healthcare, ILL, USA) at X20 magnification (15 fields per well, 30 wells per condition). Following image acquisition, high-content image stacks were analyzed using the InCarta software (GE Healthcare) producing comparative fluorescence intensity measurements. Cell and nuclei morphology were determined by cytoplasmic and nuclear staining using Calcein Red-Orange and Hoechst (Thermo Fisher Scientific, MA, USA), respectively. In order to monitor the mitochondrial content and activity, the mitochondria were labeled with MitoTracker Green FM and MitoTracker Deep-Red FM (Thermo Fisher Scientific) which provide measurement of the mitochondrial mass and activity at the cellular level [24–26]. To assess the mitochondrial oxidative damage, the cells were stained with MitoSOX (Thermo Fisher Scientific, MA, USA), Calcein Green and Hoechst, as previously reported [27, 28]. The results obtained from the InCarta software were further analyzed by utilizing a custom python script developed at the Blavatnik center for drug discovery at Tel-Aviv university, Israel. Following Z-score-based normalization, the quantification of Mito Activity was done by dividing the MitoTracker Deep- Red intensity with MitoTracker Green intensity which was normalized with the cell intensity. Mito Fraction quantification was done by dividing the parameter of mitochondria organelles count of MitoTracker Deep-Red with MitoTracker Green (S1 Fig).

## Statistical analysis

SPSS (IBM® SPSS® Statistics; Version 22) was used for statistical analysis. All variables are expressed as means ± standard error of the mean (SEM). One-way ANOVA followed by Tukey's post-hoc test was used to compare the three groups (sham, AKI-1d, AKI-1w) unless otherwise specified. In the metabolomics analysis, metabolite intensities were normalized to cardiac mass (cardiac samples) or to sample volume (plasma samples) and analyzed using one-way ANOVA following a multiple correction step (FDR step-up) using Partek Genomics Suite 7.0. The comparison included control samples, 1 day and 1 week time point samples for either heart, or plasma samples, separately. Metabolites were considered differential when detected in both groups, and had a linear |Fold-Change|>1.5, and FDR <0.05. For the specific analysis of the metabolites affiliated with the TRP catabolism pathway (Fig 2), the differences in raw intensities of these metabolites among the 3 groups (sham, AKI-1d, AKI-1w) were analyzed by ANOVA. For the AMI study, two-way paired- Student's T-test was utilized in analyzing the differences between day 1 and day 30 in each treatment group. In all tests, $p< 0.05$ was considered statistically significant. Significance was at $p<0.05$ (*$p<0.05$; $p<0.01$;***$p<0.001$).

## Results

### Model validation

In order to validate that 5/6 nephrectomy results in renal dysfunction in the acute phase, the levels of plasma creatinine and urea obtained from the complete metabolomics analysis were compared between the three experimental groups. The data indicate that plasma creatinine was significantly elevated both on day 1 (1790 ± 4.7) and day 7 (1018 ± 94.0) post AKI induction relative to vehicle-control animals (257 ± 4.7); $p< 0.001$, Fig 1A.I. Likewise, the levels of urea were markedly increased in the plasma from 156.1 ± 27.2 in control to 468.0 ± 70.0 and

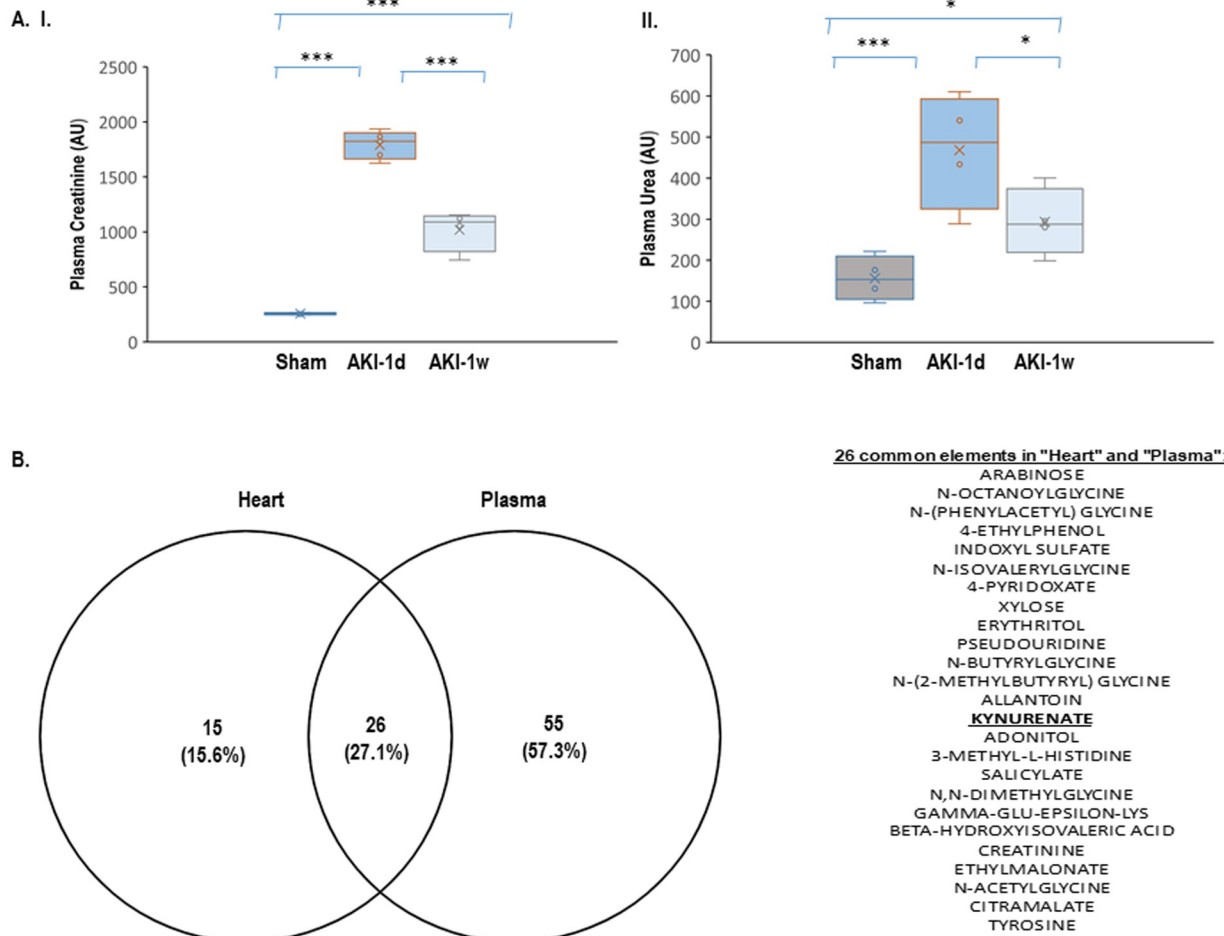

**Fig 1.** (A) Creatine and urea levels provided by the metabolomics profile of the plasma (raw data). (B) Venn diagram yielding 26 differentially-expressed cardiac and plasma metabolites at both time points: 1 day and 1 week relative to sham.

293.7 ± 41.4, on day 1 and day 7, respectively, following AKI (p<0.05, Fig 1A.II). The data point that an acute renal dysfunction takes place, shortly after AKI induction by 5/6 nephrectomy.

### Metabolomics analysis

The metabolomics analysis identified a total of 468 metabolites in the heart, which include both polar and semi-polar metabolites (276 metabolites) as well as lipid metabolites (192 metabolites) (S1 Table). Out of which, 71 and 68 metabolites were differentially expressed in the heart at 24 h and one week time points, respectively, relative to sham (FC $\geq$ 1.5; one-way ANOVA, p$\leq$ 0.05). Forty-one of these metabolites were differentially expressed at both time points relative to sham (listed as "Differential" in S1 Table). In the plasma, 440 metabolites which include both polar and semi-polar metabolites (245 metabolites) as well as lipid metabolites (195 metabolites) were identified (S2 Table). Out of which, 125 and 104 metabolites were differentially expressed at 24 h or 1 week, respectively, compared to sham (FC $\geq$ 1.5; one-way

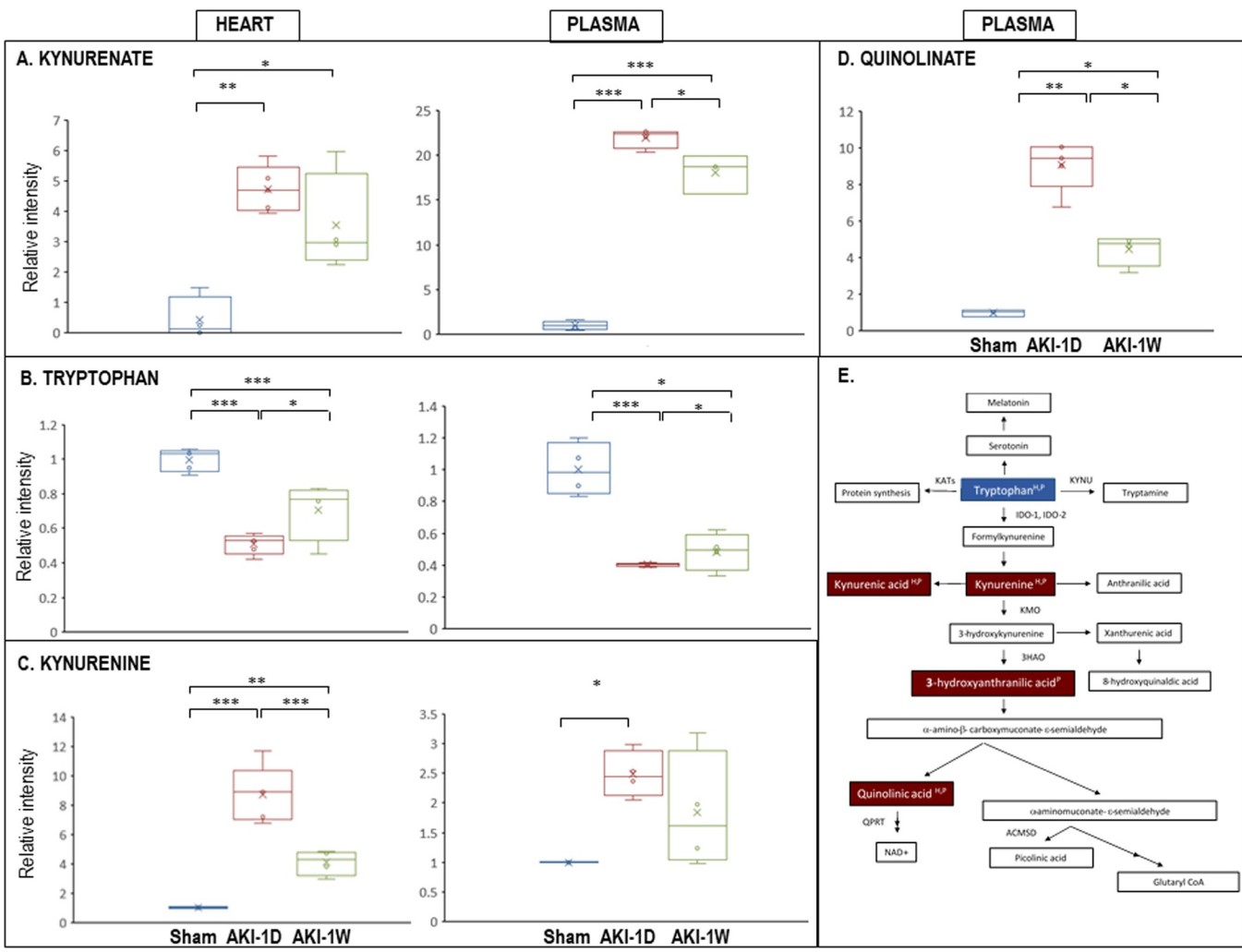

**Fig 2. AKI significantly affects the expression of various metabolites related to TRP catabolism.** (A-D) Box plot graphs showing the change of relative intensity in AKI-1d (n = 5) and AKI-1w (n = 4) versus the mean intensity value of the sham group (n = 5) per each metabolite. Significance was calculated by one-way ANOVA followed by Tukey's post-hoc test. X- Denotes mean values, horizontal lines stands for the median values. (E) Schematic illustration of the TRP catabolite pathway demonstrating increased (red), decreased (blue) or unchanged (white) expression in AKI heart (H) and plasma (P) sections at 24 hours versus sham.

ANOVA, $p \leq 0.05$). Eighty-one of these metabolites were differentially expressed at both time points relative to sham (listed as "Differential" in S2 Table). A Venn diagram analysis between the 41 cardiac common metabolites and the 81 plasma common metabolites revealed 26 over-lapping metabolites whose expression was up/down-regulated relative to sham (Fig 1B). Out of which, we chose to focus on kynurenate (kynurenic acid, KYNA)- a side-chain product of the TRP metabolism, highly secreted to the blood following AKI [29, 30] that may also be involved in cardiovascular physiology and pathology [29, 31, 32] (. An ANOVA-analysis-based fold-change in the cardiac and plasma levels of the TRP pathway metabolites relative to sham is given in Fig 2. Cardiac KYNA levels were increased by 4.4-fold on day 1 and by 3.1-fold on day 7 following AKI induction relative to sham animals. Likewise, in the plasma, KYNA levels were elevated by 22.6- fold and 17.8-fold on the same two time points, respectively ($p< 0.05$; Fig 2A). Concomitantly, TRP levels were reduced by 2.0-fold and 1.5-fold in the LV samples on day 1 and day 7, respectively ($p< 0.05$), and by 2.7 and 2.1-fold in their

plasma samples (p< 0.05; Fig 2B). Further analysis confirmed that kynurenine (KYN) -the major source key product of the TRP catabolism pathway, is also increased in the cardiac lysates and in the plasma of the AKI animals at both time points (8.7 and 4.1-fold increase in the heart, 2.7 and 2.0-fold increase in the plasma on 1 day and day 7, respectively, p< 0.05; Fig 2C). A similar increase was measured in the levels of a downstream product of the KYN pathway, Quinolinate, that was detected in cardiac samples on day 1 and day 7 post AKI, but not in the sham animals and therefore the exact fold-change could not be determined. A marked increase of this metabolite was detected also in the plasma samples on day 1 and day 7 relative to sham; (9.0-fold and 4.4-fold, respectively, p< 0.01; Fig 2D). Altogether, the results suggest that the TRP metabolism pathway in general and the KYNA metabolite, in particular, may play an important role in the pathological changes that take place in the heart in the AKI setting (Fig 2E). The raw data for KYNA, TRP and KYN induction in the cardiac lysates and the plasma samples relative to sham-operated controls are given in S2 Fig.

### Cardiac cell protection

To assess the potential effects of KYNA on the viability of cardiac cells, we first tested the viability of H9C2 myoblast cells in the presence of increasing concentrations of KYNA by flow cytometry with annexin-FITC-PI, as demonstrated in Fig 3A. As expected, in the absence of KYNA, the viability of the cells was significantly reduced under anoxia (60.3 ± 3.7%) compared to normoxia (91.7 ± 0.4%) (p = 0.01). KYNA administered at a concentration of 5 mM or 10 mM resulted in a minor reduction in the viability of the cells grown under regular normoxic conditions (86.2 ± 0.7% and 82.4 ± 1.3%, respectively; p<0.05). Surprisingly, however, preincubation with 5 mM or 10 mM KYNA significantly protected the cardiac cells from a proceeding anoxic damage (77.5 ± 1.9% and 84.2 ± 1.0% in the presence of 5 mM or 10 mM KYNA, respectively; p< 0.05). H9C2 cell viability was not affected at all when KYNA was administered at lower doses, neither under normoxia, nor under anoxia relative to control cells grown without KYNA. In view of the observed dramatic effect of KYNA when given at 10 mM concentration, we chose to use this dose in all our subsequent *in vitro* experiments. Representing flow cytometry captures demonstrating the viability of H9C2 cells grown with or without KYNA (10 mM) followed by exposure to normoxia or anoxia are given (Fig 3B). Altogether, the data indicate that while KYNA given at a concentration of 10 mM slightly minimizes the viability of H9C2 under normoxic conditions, the metabolite salvages the cells from anoxia-related cell death.

### GPR35 receptor internalization

The flow cytometry data presented herein (Fig 3C) demonstrate that anoxic conditions result in a significant increase in GPR35 surface expression relative to normoxia from 9.5% to 99.7% (P2+P3 gates), as previously suggested [14]. Exposure to KYNA (10 mM) leads to reduced GPR35 expression from the cell surface under normoxic conditions (1.0%- P2 only) as well as under anoxic conditions (16.9% versus 0.0% in the P3 gate which represents the highest-GPR35 expressing cells), probably due to receptor internalization as previously reported [33]. The data suggest that the KYNA-GPR35 interaction may initialize an intracellular signaling cascade that results in H9C2 cell salvage upon exposure to anoxia.

### Protection from anoxia-driven mitochondrial damage

H9C2 cells were grown under normoxia or anoxia, with or without prior exposure to KYNA (10 mM). The cells were labled with Hoechst (nuclei), Calcein Red-Orange (cytoplasm), Mito-Tracker Green (FITC), FM, which is recognized to represent mitochondrial mass, and

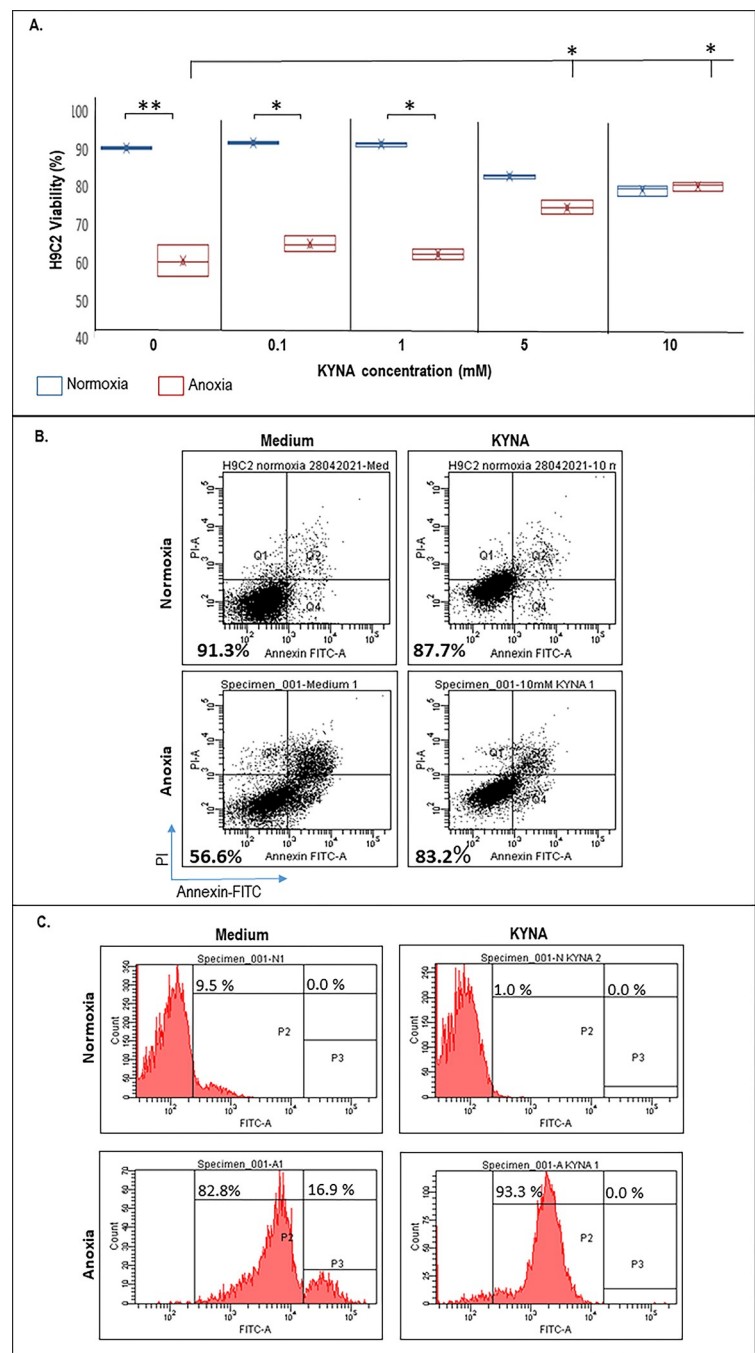

**Fig 3. KYNA salvages H9C2 cell viability upon exposure to ischemic conditions.** (A) Dose response for H9C2 cell viability grown in the presence of KYNA followed by exposure to normoxic or anoxic conditions, as determined by flow cytometry. Each box represents values derived from three independent experiments. X- Denotes mean values, horizontal lines stands for the median values. (B) Representetative flow cytometry captures showing H9C2 viability. (C) Reduced cell surface expression of GPR35 following exposure to KYNA, determind by flow cytometry (two independent experiments with two technical replicates).

MitoTracker Deep Red FM- a known mitochondrial potential-dependent indicator that reflects the mitochondrial function [24–26]. Principal Component Analysis (PCA) analysis confirmed that the combination of mitochondrial features capture 73% of the total variance in

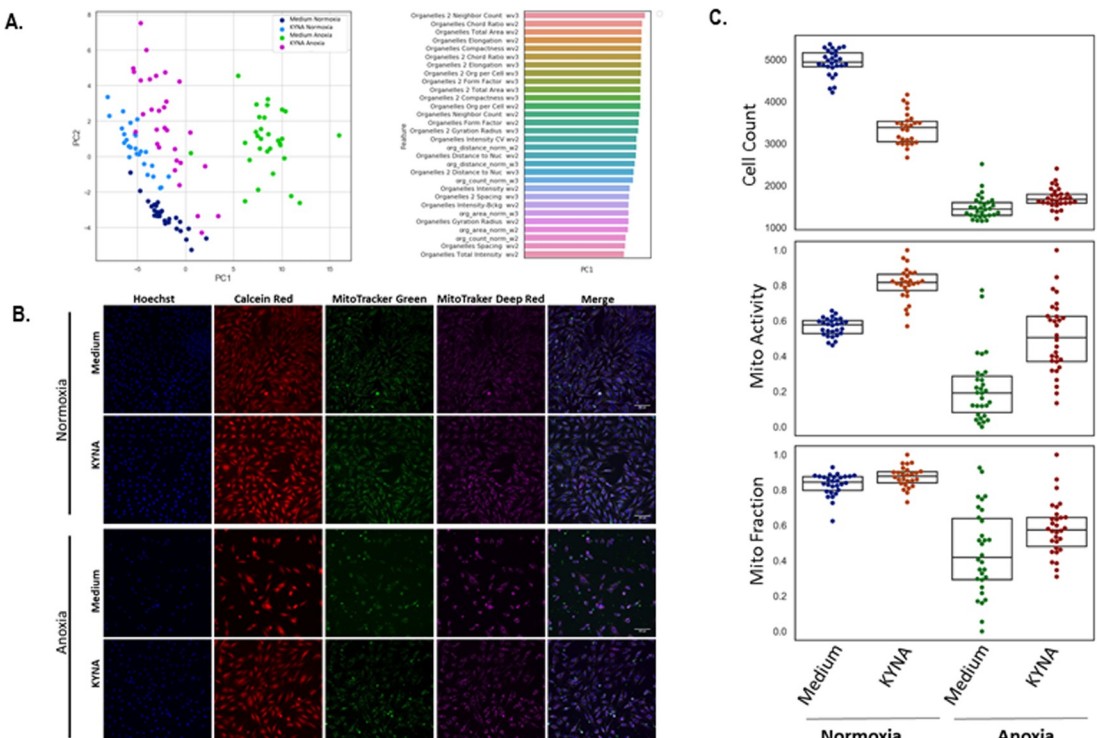

**Fig 4. KYNA rescues H9C2 cells from anoxia-driven cell death and mitochondrial disruption.** (A) A PCA analysis demonstrating a clear separation between experimental conditions by the combination of mitochondrial features (24–30 wells per treatment). (B) Representative images of mitochondrial content (MitoTracker Green FM) and function (MitoTracker Deep Red FM) as well as nuclei (Hoechst) and cytoplasm (Calcein-Red-Orange) staining. The merged column represents the merging of Hoechst and the two MitoTracker dyes. (C) Box plot graphs demonstrating median and sample scattering of the values (24–30 wells per treatment) affiliated cell count, mitochondrial fraction and mitochondrial activity after Z-score-based normalization.

the data and displays a clear separation between the experimental conditions: normoxia-medium, normoxia-KYNA, anoxia-medium and anoxia- KYNA (Fig 4A). Representative captures for single stainings and merged staining for each experimental arm as well the box plot analyses are given in Fig 4B & 4C. As expected, relative to standard normoxic conditions, exposure to anoxia significantly reduces the total cell count (1442.00 [1293.50–1596.25] vs 4937.50 [4827.5–5163.25]; $p = 3.6*10^{-45}$), the mitochondrial fraction (0.419 [0.293–0.639] vs 0.845 [0.798–0.876]; $p = 2.6*10^{-29}$) and the mitochondrial function (0.192 [0.081–0.287] vs 0.578 [0.529–0.601]; $p = 2.3*10^{-20}$). In the presence of KYNA, the stainings confirm that this metabolite reduces the cell number under physiological conditions (3382.5 [3042.50–3529.50] vs 4937.6 [4827.50–5163.25]; $p = 1.3*10^{-22}$, with or without KYNA respectively); however, KYNA slightly increases the cell number upon exposure to anoxia (1670 [1584.00–1795.00] vs 1442 [1293.50–1596.25]; $p = 0.002$).

Interestingly, staining with the two MitoTracker dyes, demonstrated that KYNA enhanced the mitochondrial content (MitoTracker Green: 0.879 [0.841–0.904] vs 0.845 [0.798–0.876]; $p = 6.2*10^{-11}$) and function (MitoTracker Deep-Red: 0.818 [0.772–0.864] vs 0.578 [0.529–0.601]; $p = 3.65*10^{-22}$) under normoxia. A more pronounced mitochondrial protective effect was evident when KYNA was administerd prior to exposure to anoxia: MitoTracker Green: 0.575 [0.481–0.645] vs 0.419 [0.293–0.639]; $p = 1.56*10^{-18}$ and MitoTracker Red: 0.505 [0.370–0.626] vs 0.192 [0.081–0.287]; $p = 3.99*10^{-9}$. Altogether, the statistical analysis of the cell count, mitochondrial fraction and mitochondrial activity in the four groups confirms that

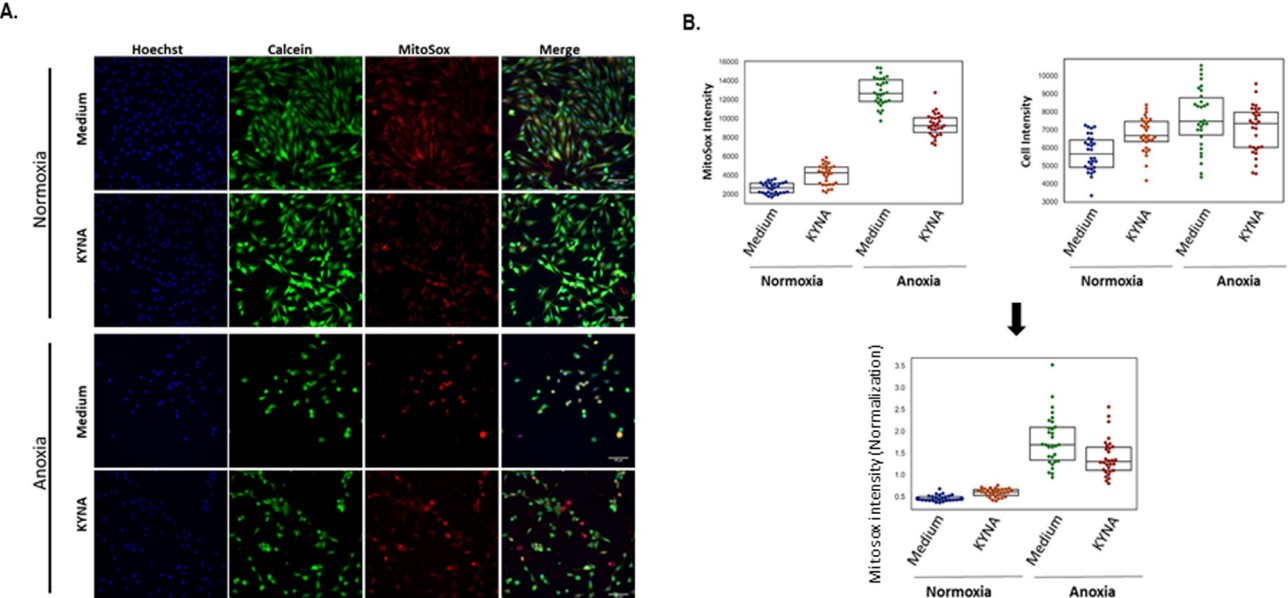

**Fig 5. KYNA reduces the anoxic-mediated oxidative damage to the mitochondria of H9C2 cells.** (A) Representative images of cytoplasm (Calcein-Green), superoxide levels (MitoSOX^TM) and nuclei (Hoechst) staining. (B) Box plots demonstrating median and sample scattering by the combination of Calcein Green and MitoSOX staining after Z-score normalization (30 wells per treatment). The statistical analysis (One-way ANOVA) is given in the text.

administration of KYNA prior to exposure to anoxia results in an increased median cell number, increased mitochondrial fraction and increased mitochondrial activity relative to cells exposed to anoxia only.

Next, since oxygen deprivation is known to stimulate mitochondrial reactive species (mROS) which could be detected in various live cells including cardiomyocytes via MitoSOX staining [27, 28], we sought to assess the potential effect of KYNA on mROS production under normoxia and anoxia. As expected, MitoSOX staining revealed a significant superoxide increase in anoxic cells relative to normoxic ones, as indicated by the representative captures (Fig 5A) and the quantitative analysis (Fig 5B). The median MitoSOX intensity was 0.455 [0.415–0.496] in normoxia versus 1.684 [1.336–2.086] in anoxia; (p = $2.4^{*}10^{-13}$). However, preincubation with KYNA, moderately reduced the superoxide levels in the anoxic cells (1.3 [1.101–1.631]; p = 0.0025 relative to anoxia only), but not in the normoxic cells-where a moderate increase was documented in the presence of the metabolite (0.602 [0.525–0.661]; p = $0.58^{*}10^{-7}$). The data indicate that KYNA may protect cardiac mitochondria from ischemic damage, at least in part, by ameliorating the anoxic-mediated oxidative damage.

## Cardiac protection in a murine model for AMI

Based on the *in vitro* data pointing to cardiac protection by KYNA upon $O_2$ restriction, and due to the known significantly increased risk for cardiac ischemic events in patients suffering from renal failure [34], we sought to assess the effects of KYNA in an *in vivo* model for AMI induced by a permanent LAD ligation. The echocardiography, presented in Fig 6A, shows that the mean LV ejection fraction (EF) in the vehicle-treated animals was reduced between day 1 and day 30 (39.9± 5.2% vs 28.3± 4.9%, p = 0.0001, two-tailed paired Student's T-test;). Interestingly, the calculated EF of the animals treated with KYNA was significantly higher on day 30 relative to day 1 (55.7± 2.0% vs 41.6 ± 3.0%, p = 0.02, two-tailed paied Student's T-test;

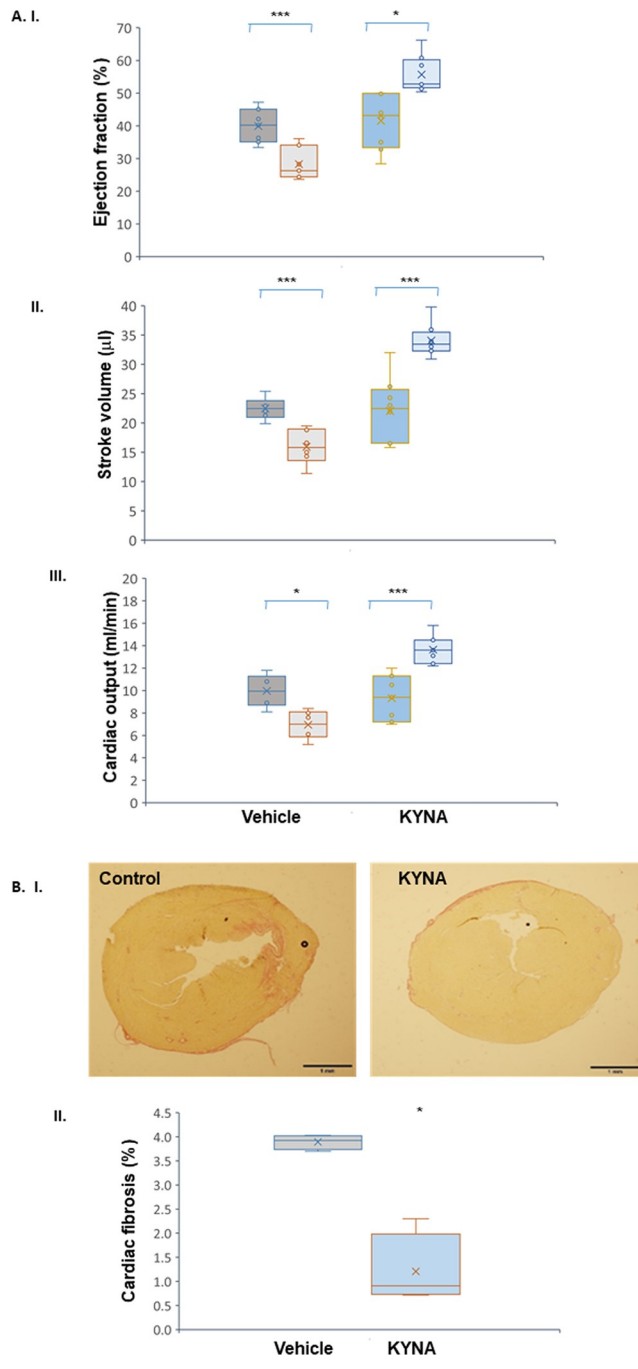

**Fig 6. KYNA enhances cardiac recovery following AMI.** Female BALB/C mice were treated with KYNA (250 mg/Lit) diluted in their drinking water (n = 9) versus regular tap water (n = 10). (A) KYNA treatment resulted in an elevated LV EF (%), SV (μl), and CO (ml/min) measurements relative to vehicle-treated control animals on day 30 relative to day 1. Statistical difference between day 1 and day 30 was determined by paired two-tailed Student's T-test. X- Denotes mean values, horizontal lines stands for the median values. (B) Concamitantly, KYNA reduced the percentage of collagen deposition in the heart according to Picro Sirius Red staining. i. representative captures of the LV sections. ii. Box plot demonstrating the collagen content out of the total LV section (n = 6/ arm). Significance was determined by two-tailed Student's T-test.

Fig 6Ai). Likewise the mean cardiac stroke volume was reduced in the control group between days 1 and 30 (22.5± 1.9 μl vs 15.9± 1.1 μl, p = 0.001) and was significantly increased in the KYNA-treatment arm (22.1± 2.0 μl vs 34.0±1.0 μl, p = 0.001; Fig 6Aii). In line with these finding the cardiac output value was reduced in the control arm (10.0± 0.6 ml/min vs 7.0± 0.5 ml/min, p = 0.03) and markedly increased following treatment with KYNA (9.3± 0.7 ml/min vs 13.6± 0.4 ml/min, p = 0.008; Fig 6Aiii) following KYNA treatment. No statistical differences were observed between the control and the KYNA groups on day 1 in all three tested cardiac parameters (p> 0.2). The histological staining for collagen content further supports the beneficial effect of KYNA on the cardiac scar size and interstitial fibrosis, as shown in the representative captures (Fig 6Bi) as well as in the quantitative analysis (4.0± 0.5% and 0.7± 1.1% in control vs KYNA arms, respectively, p = 0.03; Fig 6Bii). Altogether, the *in vivo* data strongly suggest that KYNA protects cardiac function following an ischemic event, compared to post-AMI animals which are not treated with this metabolite.

## Discussion

In recent years there is an increasing appreciation of the role of amino acids in general and of TRP in particular both in CKD [35] and in cardiovascular metabolism [36]. In line with the known TRP metabolism pathway and the reported data from human and rodents [35, 37], our analysis demonstrates that in both cardiac and plasma samples the metabolites' levels are: TRP > KYN> KYNA, so that TRP levels are higher than KYNA by at least 3-order of magnitude (S2 Fig). Moreover, it has been suggested that in CKD patients the plasma levels of TRP decrease from 96.3± 18.3 to 62.9 ± 22.8 μmole/L, while KYNA levels are increased from 24.2± 7.8 to 43.3 ± 18.2 nmole/L; p< 0.001) [35]. The data presented herein suggest that the TRP metabolism pathway in general and kynurenic acid, in particular, are significantly modified following AKI (Fig 2) and that KYNA may play a significant role in mediating cardiac protection from an ischemic damage (Figs 3–6). The marked elevation in TRP catabolites in the plasma concomitantly to their induced levels in cardiac sections following AKI, may, represent a compensatory mechanism aiming to ameliorate cardiac insult in case of an ischemic event following AKI. Indeed, previous reports suggest that the TRP catabolism products might hold a role in mediating cardiovascular physiology and pathology [38–40], in addition to the regulation of the cardiac immune system functions and inflammation [41, 42].

TRP metabolic pathway is activated during inflammatory conditions such as viral invasion, bacterial lipopolysaccharide or interferon-γ stimulation [43, 44]. TRP metabolism is well-controlled under physiological conditions but altered as part of the activated immune response. Indeed, dysregulated TRP catabolism has been linked to various diseases including neurodegenerative disorders, multiple sclerosis, schizophrenia, depression, diabetes, cancer, inflammatory bowel disease [45–47] and to an increased risk for AMI in patients with suspected angina pectoris [48].

Emerging evidence suggests that the reduced plasma concentrations of TRP concomitantly to the induced concentration of its metabolites may serve as vital biomarkers in the monitoring of several heart diseases, including AMI, atherosclerosis, and endothelial dysfunction as well as their risk factors [38–40]. Alteration in TRP catabolites was monitored in atherosclerosis and an elevated expression of indoleamine-2,3-dioxygenase (IDO)-1/2, the first and rate-limiting enzyme catabolizing both D and L-tryptophan to KYN was observed in the macrophage-rich cores of human advanced atherosclerotic plaques [49]. Moreover, several reports suggest that the cytokine interferon-gamma, which is released during the cell-mediated immune responses that take place in coronary heart diseases, induces IDO-1 activity [41, 42]. Likewise, the potential involvement of metabolites derived from TRP catabolism in general

and of KYN/KYNA, in particular has been demonstrated in the sera of patients with coronary heart disease (CHD). One study has shown that a significant proportion of patients with CHD present with decreased plasma TRP concentration which coincided with increased KYN/TRP ratio and also with increased neopterin concentrations indicating an activated immune response [41]. In addition, abnormal KYN/KYNA was found to be highly associated with endothelial dysfunction, which accounts for a significant portion of all cardiovascular diseases [50].

An elevated expression of metabolites derived from TRP catabolism was also reported in the plasma of patients suffering from kidney injury. One report showed an increased expression of the TRP catabolites L-KYN and quinolinic acid in the serum of both rat and human with renal insufficiency [30] and another one pointed to KYNA elevation in the sera of CKD patients, mainly in those suffering from polycystic kidney disease [29]. Likewise, recent findings suggest that the plasma concentrations of TRP and its catabolites might serve as biomarkers for the monitoring of AMI [38]. Yet, according to our knowledge, this is the first report that shows a marked elevation in TRP catabolites in the sera following AKI, concomitantly to their reduced expression in cardiac sections. In this respect it is worthwhile to note innovative observations suggesting that plasma KYNA may promote remote ischemic preconditioning by binding to GPR35 expressed on cardiomyocytes [51]. GPR35 bound to KYNA can bind to the mitochondrial protein ATP synthase inhibitory factor subunit 1 (ATP1F1), thus regulating the ATP synthase complex, leading to its subsequent dimerization. The activation of ATP synthase potentially overcomes the overall loss of ATP levels, which contributes to cell death upon ischemia. In addition, it has recently been reported that KYNA may enable an enhanced recovery and functionality of an injured ischemic/reperfusion aorta, potentially by promoting reactive oxygen (ROS)- dependent tissue repair via acting on oxidant enzymes [52].

The data presented herein demonstrate that KYNA somewhat decreased the cellular viability, induced some degradation of the mitochondria, reduced the mitochondrial membrane potential and induced marginal oxidative damage under physiological conditions, as indicated (Figs 3–5), in line with other reports [53]. Interestingly, however, under anoxic conditions, KYNA significantly increased the viability (Fig 3), reduced the clearance of damaged mitochondria, and increased the mitochondrial membrane potential and their resistance to oxidative stress in H9C2 cells (Figs 4 and 5). Moreover, KYNA enhanced cardiac recovery from an ischemic AMI *in vivo* (Fig 6). In line with these data, several reports point that KYNA is an important endogenous antioxidant and that its protective effects in diverse toxic models may stem from its redox characteristics [54, 55] in addition to its agonist activity on its putative receptor, the previously orphan G-protein coupled receptor 35. There is evidence that KYNA may modify heart function, yet it is debatable whether these effects are beneficial or detrimental. On the one hand, KYNA was reported to decrease respiratory parameters, mainly the respiratory control index of glutamate/malate respiring heart mitochondria, and thus lead to the development of cardiomyopathy symptoms [32, 53]. On the other hand, cardio-protective effects of KYNA have also been proposed. For instance, KYNA was proven successful in decreasing the heart rate of spontaneously hypertensive rats without affecting the mean arterial pressure and thus it is suggested that KYNA may offer therapeutic potential as opposed to most drugs used to decrease heart rate that have strong adverse inotropic or hypotensive effects [31]. Recent reports show that KYNA may exert anti-inflammatory effects through GPR35 internalization and activation which in turn inhibits the release of TNFα by macrophages under LPS-induced inflammatory conditions [33]. In the heart, the expression of GPR35 may correlate with heart failure. Indeed, GPR35 was identified among the top 12 genes whose expression correlates with the severity of heart failure and adenoviral overexpression of GPR35 was shown to cause hypertrophic-like morphology changes and reduced cellular

viability of cardiomyocytes [13]. Interestingly, in line with our data (Fig 3C), recent reports suggest that the plasma membrane expression of GPR35 in cardiomyocytes is significantly induced under hypoxia and that application of an alternative GPR35 ligand, KYNA or zaprinast, induces GPR35 internalization and subcellular accumulation in the myocytes [14, 56]. Furthermore, it has been suggested that an *in vivo* administration of a siRNA to GPR35 enhances cardiac function following an acute MI through reduction of reactive oxygen species activity and mitochondria-dependent apoptosis [56]. The data presented herein suggest that KYNA administration prior to the induction of an ischemic event, leads to GPR35 intracellular internalization concomitantly to reduced cell death. We thus assume that the receptor internalization or downregulated expression may protect the cardiomyocytes from cell membrane GPR35-mediated cell death and LV dysfunction. Previous studies have demonstrated that the KYNA-GPR35 interactions can inhibit N-type $Ca^{2+}$ channels in sympathetic neurons [57] and reduce the plateau phase of ATP-induced calcium transients in astrocytes [58]. Therefore it is yet to be clarified whether the elevated levels of GPR35 upon exposure to hypoxia result in increased detrimental $Ca^{2+}$-influx into the injured cardiomyocytes and whether the KYNA-mediated GPR35 internalization reduces these pathological enhanced $Ca^{2+}$ transients. Thus, in line with these reports and data from our *in vitro* and animal studies, several mechanisms can be proposed to explain the dual-effect of KYNA that can potentially be harnessed for medical purposes. KYNA was shown to have antioxidant effect, it has effect on calcium ion influx and efflux and importantly on membranous presentation of GPR35 receptor and mainly internalization of the receptor upon exposure to high levels of the metabolite.

## Study limitation

The effect of KYNA in our study was performed in a model of LAD complete ligation and not in a reperfusion injury model. Likewise, the beneficial effect of KYNA should also be validated in an AMI model in male rats, in line with the *in vitro* beneficial effect demonstrated herein for anoxic H9C2 rat myocytes and in view of the entire metabolomics analysis that was conducted in a male rat AKI model. In addition, the *in vivo* efficacy experiment involved only one dose of KYNA: 250 mg/ L which corresponds to 1.3 mM, whereas the physiological concentration of KYNA in the plasma is within the nM range. Though, the current dose was not toxic, lower doses should also be attempted in further studies. Furthermore, the mitochondrial analysis performed in H9C2 cells should be repeated in primary cardiomyocyes due to the different characteristics between the two cell types. In addition, future studies should include measurements of the actual KYNA levels in the heart and the plasma via targeted-metabolomics analysis.

## Conclusions

The data presented herein shed light on the tryptophan catabolism product, KYNA, which may represent a key metabolite absorbed by the heart following AKI. The data highly suggest that KYNA can enhance cardiac cell viability following an ischemic event both *in vitro* and *in vivo* in a mechanism which is mediated, at least in part, by the protection of the cardiac mitochondria from oxidative stress. Further research is warranted in order to determine whether KYNA can be developed into a therapeutic drug or a supplemental agent to be utilized in the multimodality treatment of post-AMI patients and in patients who are at high risk for experiencing an ischemic event, while minimizing its toxicity to cardiomyocytes under physiological conditions.

In addition, the exact mechanism of action of the KYNA-GPR35 axis which leads to cardiac protection following ischemia is yet to be deciphered.

## Supporting information

**S1 Fig. Parameters which exhibit the rescue of H9C2 cells grown under anoxia in box plots graphs.** (A) Quantification of Mito Activity and Mito fraction as documented in Fig 4. (B) Box plots graph representation of MitoTracker Deep Red intensity, MitoTracker red intensity, and cell intensity. (C) Box plots graph re-presentation of mitochondria elongation and area (24–30 replicates per arm).
(TIF)

**S2 Fig. Cardiac and plasma levels of KYNA, TRP and KYN following AKI (raw data).**
(TIF)

**S1 Table. List of all metabolites detected in cardiac lysates of the experimental animals (arbitrary units).** N/A- not applicable = below detection. Metabolite intensities of both polar and semi-polar metabolites were normalized to cardiac mass (cardiac samples) or to sample volume (plasma samples) and analyzed using ANOVA following a multiple correction step (FDR step-up), using Partek Genomics Suite 7.0. Differential metabolites whose levels are significantly modified in both 24h and 7 day- time points relative to sham are listed in the right column.
(XLSX)

**S2 Table. List of all metabolites detected in plasma samples of the experimental animals (arbitrary units).** N/A- not applicable = below detection. Metabolite intensities of both polar and semi-polar metabolites were normalized to cardiac mass (cardiac samples) or to sample volume (plasma samples) and analyzed using ANOVA following a multiple correction step (FDR step-up), using Partek Genomics Suite 7.0. Differential metabolites whose levels are significantly modified in both 24h and 7 day- time.
(XLSX)

## Acknowledgments

We thank Prof. Atan Gross, Dr. Smadar Levin-Zaidman and Dr. Nili Dezorella from the Weizmann Institute of Science for carefully reading the manuscript and their fruitful advice.

## Author Contributions

**Conceptualization:** Einat Bigelman, Michal Entin-Meer.

**Data curation:** Einat Bigelman, Michal Entin-Meer.

**Formal analysis:** Einat Bigelman, Metsada Pasmanik-Chor, Bareket Dassa, Orly Dorot, Edward Pichinuk, Michal Entin-Meer.

**Investigation:** Einat Bigelman, Michal Entin-Meer.

**Methodology:** Einat Bigelman, Maxim Itkin, Sergey Malitsky, Orly Dorot, Yuval Kleinberg.

**Resources:** Edward Pichinuk, Gad Keren.

**Software:** Metsada Pasmanik-Chor, Bareket Dassa, Michal Entin-Meer.

**Supervision:** Gad Keren, Michal Entin-Meer.

**Writing – original draft:** Einat Bigelman, Michal Entin-Meer.

**Writing – review & editing:** Gad Keren, Michal Entin-Meer.

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
