## [Decision Letter · Decision Letter 0]

6 Feb 2023

PONE-D-22-25974Kynurenic acid, a key L-tryptophan-derived metabolite, protects the heart from an ischemic damagePLOS ONE

Dear Dr. Entin-Meer,

Thank you for submitting your manuscript to PLOS ONE. After careful consideration, we feel that it has merit but does not fully meet PLOS ONE’s publication criteria as it currently stands. Therefore, we invite you to submit a revised version of the manuscript that addresses the points raised during the review process. Please have a look at all the comments brought up by the reviewers, especially regarding methodological descriptions. Also please ensure there is a clear flow of the manuscript.

We look forward to receiving your revised manuscript.

Kind regards,

Daniel M. Johnson, PhD

Academic Editor

PLOS ONE

Journal Requirements:

Reviewers' comments:

Reviewer's Responses to Questions

**Comments to the Author**

1. Is the manuscript technically sound, and do the data support the conclusions?

Reviewer #1: Partly

Reviewer #2: Yes

2. Has the statistical analysis been performed appropriately and rigorously? 

Reviewer #1: Yes

Reviewer #2: Yes

3. Have the authors made all data underlying the findings in their manuscript fully available?

Reviewer #1: Yes

Reviewer #2: Yes

4. Is the manuscript presented in an intelligible fashion and written in standard English?

Reviewer #1: Yes

Reviewer #2: Yes

5. Review Comments to the Author

Reviewer #1: The present manuscript authored by a group of investigators reported effects of kynurenic acid on cardiac cell viability or contractile function following exposure to in vitro hypoxia or in vivo myocardial ischemia. Possible involvement of cardiac mitochondria in mediating these effects was also investigated in vitro. They demonstrated an important role played by kynurenic acid as a key metabolite after acute kidney ischemia (AKI) that may paradoxically induce cardioprotection. Overall this is an interesting and elaborated study that has provided some new findings and insights. Please consider the following concerns or suggestions of mine:

1) Page 6, Line 119: Why 6 mice were excluded due to “normal cardiac function” on Day 1 after LAD ligation? Was this a surgical technical issue or due to resistance of female mice to myocardial ischemia? If so, why the authors chose female mice not male rats for AMI studies, considering the AKI studies were performed in male rats? Why used different animal species and genders?

2) Page 6, Line 130: Please correct “into 5 µM slices” with “into 5 µm slices”.

3) Page 7, Line 146 to 154: The description of statistical analysis should be removed here and merged to Page 9, Line 189 to 195, in order to avoid redundancy.

4) Page 9, Line 165 and 171: How the sole concentration of 10 mM kynurenic acid was selected for the H9C2 experiments? Is 10 mM a physiologically achievable concentration that reflects tryptophan degradation in vivo? Please justify.

5) In Discussion section, it would be helpful to refer the particular figure(s) or table(s) to justify each of your claims or reasoning, based on the results of the current study.

6) Page 19, Line 427-428: The paragraph of “Study Limitation” seems to be quite superficial and needs to be expanded with further details and/or explanations. For example, why did the authors use two different species and genders, male rats versus female mice in AKI and AMI studies? Also mitochondrial study was done only H9C2 cells, which have different cellular characteristics from primary cardiomyocytes. This should be discussed as a study limitation.

7) Do you think the release of cardioprotective substances to circulation after AKI may potentially explain the phenomenon of “remote preconditioning”? If so, please add some discussion on this possibility.

8) Page 25, Ref. 42: The journal volume/page numbers are missing for this reference.

9) Figure 6 should use scatter dot plots, instead of bar graphs, in order to reveal all individual data points. Also please report the heart rate data for all groups of mice, since previous study (Ref. 22) showed that in vivo treatment of kynurenic acid significantly reduced heart rate in rats. Please discuss if such heart rate reduction contribute to the cardioprotective effects of kynurenic acid in the present study.

Reviewer #2: 1. In Fgireu 1, the authors used 26 differentially-expressed cardiac and plasma metabolites at both time points: 1 day and 1 week relative to sham. However, the reviewer cannot understand the reason underlying the time point selection.

2. It requires to further evaluate the concentration of KYNA in the tissue and serum.

3. Although the authors reported that KYNA is able to reduces ischemia-mediated H9C2 damage, the molecualr mechanisms undelrying cell death has not been addressed.

4. The 5/6 nephrectomy is actually a chronic renal failure model but not acute kidney injury.

5. Acute hypoxia exposure cannot stimulate the model of cardiac injury after AKI.

6. PLOS authors have the option to publish the peer review history of their article (what does this mean?). If published, this will include your full peer review and any attached files.

Reviewer #1: **Yes: **Lei Xi

Reviewer #2: No

---

## [Author Response · Author response to Decision Letter 0]

23 Feb 2023

February, 2023

To: Dr. Daniel M. Johnson, Academic editor

Re: Revision to manuscript PONE-D-22-25974:

Dear Dr. Johnson,

We would like to thank the reviewers and the editor for the in depth analysis of our work and for raising several important points that needed clarification. We appreciate the time and effort expended on our behalf. We have made significant modifications to the manuscript accordingly, and out point-by-point responses are listed below.

We hope that our revised manuscript will now be found suitable for publication in PLOS ONE.

Sincerely,

Dr. Michal Entin-Meer and Prof. Gad Keren

General comments:

https://journals.plos.org/plosone/s/file?id=ba62/PLOSOne_formatting_sample_title_authors_affiliations.pdf- Done.

When you resubmit, please ensure that you provide the correct grant numbers for the awards you received for your study in the ‘Funding Information’ section. Done.

3. We note that you have included the phrase “data not shown” in your manuscript. Unfortunately, this does not meet our data sharing requirements. PLOS does not permit references to inaccessible data. We require that authors provide all relevant data within the paper, Supporting Information files, or in an acceptable, public repository. Please add a citation to support this phrase or upload the data that corresponds with these findings to a stable repository (such as Figshare or Dryad) and provide and URLs, DOIs, or accession numbers that may be used to access these data. Or, if the data are not a core part of the research being presented in your study, we ask that you remove the phrase that refers to these data. Corrected

Reviewer 1: 

1. Page 6, Line 119: Why 6 mice were excluded due to “normal cardiac function” on Day 1 after LAD ligation? Was this a surgical technical issue or due to resistance of female mice to myocardial ischemia? If so, why the authors chose female mice not male rats for AMI studies, considering the AKI studies were performed in male rats? Why used different animal species and genders?

"Normal cardiac function" on day 1 post AMI induction means that the LAD ligation was technically unsuccessful (i.e., MI was not induced in these animals), therefore these animals were excluded from the experiment. In order to clarify the issue, we modified this sentence under "Materials & Methods" which now reads: Five animals from the control group and six animals from the experimental group were excluded from the analysis due to one of the following reasons: normal cardiac function according to echocardiography applied on day 1, suggesting that LAD ligation was unsuccessful (6 animals), animal's death prior to endpoint (4 animals) or weight loss of > 15% (1 animal) (lines 119-123).

The strain and gender for each experimental procedure (AKI and AMI) was chosen based on practical reasons per se. We added an explanation as follows: "Male Lewis rats (250–350 gr)/ 12-week old female BALB/C mice were purchased from Envigo, Israel. Male Lewis rats were used for AKI studies, as previously established in our laboratory for 5/6 nephrectomy models (9). Female mice were utilized in the AMI studies, in accordance with the extensive experience gathered in our laboratory (15, 16) and the lower cardiac rupture rate of female compared to male mice following AMI induction (17) (lines 95-99). We agree that this issue may be confusing, therefore we also added the following sentence under "Study limitation: "Likewise, the beneficial effect of KYNA should also be validated in an AMI model in male rats, in line with the in vitro beneficial effect demonstrated herein for anoxic H9C2 rat myocytes and in view of the entire metabolomics analysis that was conducted in a male rat AKI model" (lines 456-459).

2. Page 6, Line 130 (currently 134): Please correct “into 5 µM slices” with “into 5 µm slices”-done.

3. Page 7, Line 146 to 154: The description of statistical analysis should be removed here and merged to Page 9, Line 189 to 195, in order to avoid redundancy

Corrected. The explanation regarding the statistical analysis is currently merged (lines 184-196).

4. Page 9, Line 165 and 171: How the sole concentration of 10 mM kynurenic acid was selected for the H9C2 experiments? Is 10 mM a physiologically achievable concentration that reflects tryptophan degradation in vivo? Please justify.

As the reviewer noted, when reading the "Materials & Methods" section which precedes the Results section, it is unclear why 10 mM dose of KYNA was used in the in vitro experiments. Therefore, we omitted the dose from "Materials and Methods", while further clarifying the issue in the Results as explained in lines 266-266: “In view of the observed dramatic effect of KYNA when given at 10 mM concentration, we chose to use this dose in all our subsequent in vitro experiments”, as also shown in Figure 3A. The 10 mM dose was then mentioned in the description of the subsequent in vitro experiments and the observed results (lines: 267, 274, 287).

5. In Discussion section, it would be helpful to refer the particular figure(s) or table(s) to justify each of your claims or reasoning, based on the results of the current study- Done.

6. Page 19, Line 427-428: The paragraph of “Study Limitation” seems to be quite superficial and needs to be expanded with further details and/or explanations. For example, why did the authors use two different species and genders, male rats versus female mice in AKI and AMI studies? Also mitochondrial study was done only H9C2 cells, which have different cellular characteristics from primary cardiomyocytes. This should be discussed as a study limitation. 

 We thank the reviewer for this important comment. The suggested remarks were added to "Study limitations (lines 456-465).

7. Do you think the release of cardioprotective substances to circulation after AKI may potentially explain the phenomenon of “remote preconditioning”? If so, please add some discussion on this possibility.

 We thank the reviewer for raising this important issue. We indeed believe that the excess release of KYNA to the plasma following AKI may represent (at least in part) remote ischemic preconditioning. A newly-released Science paper as well as additional recent study point to this phenomenon. We added the data to the Discussion: "In this respect it is worthwhile to note innovative observations suggesting that plasma KYNA may promote remote ischemic preconditioning by binding to GPR35 expressed on cardiomyocytes (45). GPR35 bound to KYNA can bind to the mitochondrial protein ATP synthase inhibitory factor subunit 1 (ATP1F1), thus regulating the ATP synthase complex, leading to its subsequent dimerization (44). The activation of ATP synthase potentially overcomes the overall loss of ATP levels, which contributes to cell death upon ischemia. In addition, it has recently been reported that KYNA may enable an enhanced recovery and functionality of an injured ischemic/reperfusion aorta, potentially by promoting reactive oxygen (ROS)- dependent tissue repair via acting on oxidant enzymes (46)"(lines 407-415).

8. Page 25, Ref. 42: The journal volume/page numbers are missing for this reference- corrected.

9. Figure 6 should use scatter dot plots, instead of bar graphs, in order to reveal all individual data points. Also please report the heart rate data for all groups of mice, since previous study (Ref. 22) showed that in vivo treatment of kynurenic acid significantly reduced heart rate in rats. Please discuss if such heart rate reduction contribute to the cardioprotective effects of kynurenic acid in the present study.

 We rearranged Figure 6 so that it currently uses box plots (instead of bar graphs).

 The heart rate in all echocardiographic analyses was set to 400-500 bpm, prior to taking the measurements. Due to the isoflurane anesthesia and the chest pressure inflicted by the catheter, we could not achieve a heart rate of over 500 bpm. However, a heart rate between 400-500 bpm is actually the standard of our echocardiography measurements, as presented in our earlier PLOS ONE papers (Entin-Meer et al, 2017, Bigelman et al, 2018, Cohen et al, 2020). It is also considered an acceptable rate according to Lindsey et al., Am J Physiol Heart Circ Physiol, 2018. We added this clarification under "Material and Methods-Echocardiography" as follows: "Cardiac parameters on day one and day 30 post-infarction were obtained by echocardiography (Vevo 2100, VisualSonics, Toronto, Canada), while heart rate was kept between 400-500 bpm in accordance with the guidelines for echocardiographic measurements in the murine heart (19)" (lines 125-128).

Reviewer 2:

1. In Figure 1, the authors used 26 differentially-expressed cardiac and plasma metabolites at both time points: 1 day and 1 week relative to sham. However, the reviewer cannot understand the reason underlying the time point selection.

 We chose two time points in AKI- 1d and 7d, which may represent potentially-transient vs potentially-persistent AKI. We added this explanation in lines 112-113 ("Materials & Methods"). Only the metabolites which were modified at these two time points relative to control were considered significant in our metabolomics analysis (Figure 1, Figure 2).

2. It requires to further evaluate the concentration of KYNA in the tissue and serum.

 We added Fig S4 (supplementary material) showing the raw data obtained for KYNA, KYN and TRP in the global metabolomics analysis (AU). As expected according to accumulated published data, we could see that the metabolites' levels both in the blood and the cardiac sections are as follows: TRP> KYN> KYNA. We added the following explanation to the beginning of the Discussion (lines 366-371): "In line with the known TRP metabolism pathway and the reported data from human and rodents (29, 31), our analysis demonstrates that in both cardiac and plasma samples the metabolites' levels are: TRP > KYN> KYNA, so that TRP levels are higher than KYNA by at least 3-order of magnitude (S4 Fig). Moreover, it has been suggested that in CKD patients the plasma levels of TRP decrease from 96.3± 18.3 to 62.9 ± 22.8 �mole/L, while KYNA levels are increased from 24.2± 7.8 to 43.3 ± 18.2 nmole/L; p< 0.001) (29). Yet, we agree that the actual concentration of KYNA should be evaluated via targeted-metabolimcs analysis (as opposed to the global analysis presented in the current manuscript). We added the relevant statement to "Study limitation" as follows: "In addition, future studies should include the measurements of the actual KYNA levels in the heart and plasma via targeted- metabolomics analysis" (lines 463-465).

3. Although the authors reported that KYNA is able to reduce ischemia-mediated H9C2 damage, the molecular mechanisms underlying cell death has not been addressed.

In accordance with a recently-published Science paper (Wyant et al, Mitochondrial remodeling and ischemic protection by G protein-coupled receptor 35 agonists), as well as by Lima et al, 2021- we added a paragraph to the discussion that describes the potential molecular/intracellular mechanisms underlying the cell death rescue by KYNA (lines 407-415): " In this respect it is worthwhile to note innovative observations suggesting that plasma KYNA may promote remote ischemic preconditioning by binding to GPR35 expressed on cardiomyocytes (45). GPR35 bound to KYNA can bind to the mitochondrial protein ATP synthase inhibitory factor subunit 1 (ATP1F1), thus regulating the ATP synthase complex leading to its subsequent dimerization. The activation of ATP synthase potentially overcomes the overall loss of ATP levels, which contributes to cell death upon ischemia. In addition, it has recently been reported that KYNA may enable an enhanced recovery and functionality of an injured ischemic/reperfusion aorta, potentially by promoting reactive oxygen (ROS)- dependent tissue repair via acting on oxidant enzymes (46)" In addition, we have demonstrated that the ischemia-mediated cell-death salvaged by KYNA is partly achieved by reducing the oxidative stress (Figure 5).

4. The 5/6 nephrectomy is actually a chronic renal failure model but not acute kidney injury.

We agree that 5/6 nephrectomy is a known model for CKD. Yet, the metabolomics analysis performed on the plasma samples of the experimental animals revealed a significant increase in both creatinine and urea levels relative to sham-operated control animals, implying to renal dysfunction occurring as early as 1d post model induction. We added box plots showing these data to Figure 1 (Fig1A) and a paragraph regarding model validation in the beginning of the Results section (lines 202-209).

5. Acute hypoxia exposure cannot simulate the model of cardiac injury after AKI.

We agree. Nevertheless, since AMI is an unfortunate prevalent phenomenon following renal injury, we thought to assess the potential effect of KYNA should an ischemic event occurs. We added this statement on line 334-335 as follows: "Based on the in vitro data pointing to cardiac protection by KYNA upon O2 restriction, and due to the known significantly increased risk for cardiac ischemic events in patients suffering from renal failure (28), we sought to assess the effects of KYNA in an in vivo model for AMI induced by a permanent LAD ligation." In addition, it is worthwhile to mention that an exposure of cardiac cells to anoxic/hypoxic conditions was already utilized by our group to simulate cardiac infarction (though obviously not necessarily due to AKI) as reported in Entin-Meer et al, PLoS One, 2014.

We hope that you will find the corrected manuscript entitled: "Kynurenic acid, a key L-tryptophan-derived metabolite, protects the heart from an ischemic damage" interesting for the readers of PLoS ONE.

Sincerely,

Dr. Michal Entin-Meer

---

## [Decision Letter · Decision Letter 1]

2 May 2023

PONE-D-22-25974R1Kynurenic acid, a key L-tryptophan-derived metabolite, protects the heart from an ischemic damagePLOS ONE

Dear Dr. Entin-Meer,

Thank you for submitting your manuscript to PLOS ONE. After careful consideration, we feel that it has merit but does not fully meet PLOS ONE’s publication criteria as it currently stands. Therefore, we invite you to submit a revised version of the manuscript that addresses the points raised during the review process.

Although myself and the Reviewers appreciated your response to the previous comments, there is some further comments regarding the use of the different dyes for mitochondrial structure and for mitochondrial function. Therefore, there should be some discussion regarding the use of these different dyes in a revised version of the manuscript. Additional information regarding this can be found in the comments of Reviewer 3.

We look forward to receiving your revised manuscript.

Kind regards,

Daniel M. Johnson, PhD

Academic Editor

PLOS ONE

Journal Requirements:

Reviewers' comments:

Reviewer's Responses to Questions

**Comments to the Author**

1. If the authors have adequately addressed your comments raised in a previous round of review and you feel that this manuscript is now acceptable for publication, you may indicate that here to bypass the “Comments to the Author” section, enter your conflict of interest statement in the “Confidential to Editor” section, and submit your "Accept" recommendation.

Reviewer #1: All comments have been addressed

Reviewer #3: (No Response)

2. Is the manuscript technically sound, and do the data support the conclusions?

Reviewer #1: Yes

Reviewer #3: Yes

3. Has the statistical analysis been performed appropriately and rigorously? 

Reviewer #1: Yes

Reviewer #3: Yes

4. Have the authors made all data underlying the findings in their manuscript fully available?

Reviewer #1: Yes

Reviewer #3: Yes

5. Is the manuscript presented in an intelligible fashion and written in standard English?

Reviewer #1: Yes

Reviewer #3: Yes

6. Review Comments to the Author

Reviewer #1: In this revised manuscript, the authors have carefully answered all my previous concerns and suggestions. I am glad to see the substantial improvement and have no further concern.

Reviewer #3: The manuscript by Bigelman et al. reports the identification by metabolic profiling of kynurenic acid as a differentially expressed metabolite in cardiac lysates and plasma samples of male rats subjected to acute kidney injury (5/6 nephrectomy). The authors show that kynurenic acid enhanced cardiac recovery after acute myocardial infarction (permanent LAD artery ligation) in female mice, as determined by histology and echocardiography, and rescued viability and mitochondrial structure and function in H9c2 cardiac cells grown in anoxia. Even though there are some issues related to the methodology such as the different species and genders used for the studies or the supra-physiological kynurenic acid dose administered in in vivo experiments, in general the methodology is adequate. My main concern relates to the use of MitoTracker Green for mitochondrial structure and MitoTracker Deep Red for mitochondrial function (page 13, line 288-289). How do the authors justify this? MitoTracker fluorescent probes are useful for mitochondria localization. Also, an increase in superoxide production in anoxia, as the increase reported in H9c2 cells, seems counterintuitive, since superoxide is generated from oxygen. How did the authors achieved anoxia, i.e. 0% oxygen? Even though the authors did not explore the molecular mechanisms for kynurenic acid protection, but they propose the one described by Wyant et al. (Science, 377: 621-629, 2022), the results of the study contribute to reinforce the cardioprotective role of kynurenic acid against ischemic damage. Most questions of the original reviewers have been answered in a reasonable manner. Taking all of the above into consideration, and with the caveats stated above, the manuscript is informative and relevant in the field of acute kidney injury and associated ischemic cardiac events.

7. PLOS authors have the option to publish the peer review history of their article (what does this mean?). If published, this will include your full peer review and any attached files.

Reviewer #1: **Yes: **Lei Xi

Reviewer #3: No

---

## [Author Response · Author response to Decision Letter 1]

16 May 2023

May 16th, 2023

To: Dr. Daniel M. Johnson, Academic editor

Re: Revision to manuscript PONE-D-22-25974R1:

Dear Dr. Johnson,

We would like to thank the reviewers and the editor for the in depth analysis of our work and for raising an additional point that needed clarification. We appreciate the time and effort expended on our behalf. We have modified to the manuscript accordingly, as listed below.

Reviewer #3: My main concern relates to the use of MitoTracker Green for mitochondrial structure and MitoTracker Deep Red for mitochondrial function (page 13, line 288-289). How do the authors justify this? MitoTracker fluorescent probes are useful for mitochondria localization. Also, an increase in superoxide production in anoxia, as the increase reported in H9C2 cells, seems counterintuitive, since superoxide is generated from oxygen. How did the authors achieved anoxia, i.e. 0% oxygen

Response: We thank the reviewer for raising this issue. We agree that the term "Mitochondrial structure" with respect to MitoTracker Green is not accurate and should be changed to "Mitochondrial content/mass". Yet, even though fluorescent MitoTracker probes are successfully utilized for mitochondrial localization/co-localization experiments as the reviewer rightfully noted, they could also be applied as a useful tool in evaluating mitophagy (MitoTracker Green which selectively accumulates at the mitochondrial matrix) and alterations in the mitochondrial membrane potential (MitoTracker Red whose levels of accumulation in the mitochondria depend on the membrane potential) in live cells (added Ref 24-26).

As for the MitoSOX staining: it is reported to successfully detect mitochondrial ROS production due to oxygen deprivation in various cells including cardiomyocytes, as previously reported (added Ref 27, 28). 

In view of the above-mentioned:

1. We changed the term "Mitochondrial structure" to "Mitochondrial content" or "Mitochondrial mass" when discussing the data obtained with MitoTracker Green.

2. We added several references to justify the use of the MitoTracker dyes for the assessment of mitochondrial content and activity (Ref 24-26) as well as the use of MitoSOX dye for the assessment of ROS production (Ref 27, 28) in live H9C2 cells following oxygen deprivation.

3. We added clarification regarding the utilization of the dyes to "Materials & Methods (lines 175-179) and "Results" (lines 291-292 and 315-317). We also added the following sentences under Discussion (lines 422-427): "The data presented herein demonstrate that KYNA somewhat decreased the cellular viability, induced some degradation of the mitochondria, reduced the mitochondrial membrane potential and induced marginal oxidative damage under physiological conditions, as indicated (Fig 3-5), in line with other reports (52). Interestingly, however, under anoxic conditions, KYNA significantly increased the viability (Fig 3), reduced the clearance of damaged mitochondria, and increased the mitochondrial membrane potential and their resistance to oxidative stress in H9C2 cells (Fig 4-5).

4. Per the reviewer's request, we provided more details regarding the anoxia methodology (lines 155-156 in "Materials & Methods"), and- as listed above, two references pointing that oxygen deprivation can indeed lead to MitoSOX accumulation (27, 28).

We declare that the funders had no role in study design, data collection and analysis, decision to publish, or preparation of the manuscript.

We hope that our revised manuscript will now be found suitable for publication in PLOS ONE.

Sincerely,

Dr. Michal Entin-Meer and Prof. Gad Keren

---

## [Decision Letter · Decision Letter 2]

20 Jun 2023

Kynurenic acid, a key L-tryptophan-derived metabolite, protects the heart from an ischemic damage

PONE-D-22-25974R2

Dear Dr. Entin-Meer,

We’re pleased to inform you that your manuscript has been judged scientifically suitable for publication and will be formally accepted for publication once it meets all outstanding technical requirements.

Kind regards,

Daniel M. Johnson, PhD

Academic Editor

PLOS ONE

Additional Editor Comments (optional):

Reviewers' comments:

Reviewer's Responses to Questions

**Comments to the Author**

1. If the authors have adequately addressed your comments raised in a previous round of review and you feel that this manuscript is now acceptable for publication, you may indicate that here to bypass the “Comments to the Author” section, enter your conflict of interest statement in the “Confidential to Editor” section, and submit your "Accept" recommendation.

Reviewer #3: All comments have been addressed

2. Is the manuscript technically sound, and do the data support the conclusions?

Reviewer #3: Yes

3. Has the statistical analysis been performed appropriately and rigorously? 

Reviewer #3: Yes

4. Have the authors made all data underlying the findings in their manuscript fully available?

Reviewer #3: Yes

5. Is the manuscript presented in an intelligible fashion and written in standard English?

Reviewer #3: Yes

6. Review Comments to the Author

Reviewer #3: The authors have made some modifications and improvements to the manuscript and I have no further comments.

7. PLOS authors have the option to publish the peer review history of their article (what does this mean?). If published, this will include your full peer review and any attached files.

Reviewer #3: No

---

## [Editor Report · Acceptance letter]

14 Aug 2023

PONE-D-22-25974R2 

Kynurenic acid, a key L-tryptophan-derived metabolite, protects the heart from an ischemic damage 

Dear Dr. Entin-Meer:

I'm pleased to inform you that your manuscript has been deemed suitable for publication in PLOS ONE. Congratulations! Your manuscript is now with our production department. 

Kind regards, 

on behalf of

Dr. Daniel M. Johnson 

Academic Editor

PLOS ONE